# Stretched polar vortex increases mid-latitude climate variability during the Last Glacial Maximum

Yurui Zhang[1], Hans Renssen[2], Heikki Seppä[3], Zhen Li[1], Xingrui Li[1]

[1] State Key Laboratory of Marine Environmental Science, College of Ocean & Earth Sciences, Xiamen University, Xiamen, China
[2] Department of Natural Sciences and Environmental Health, University of South-Eastern Norway, Bø, Norway
[3] Department of Geosciences and Geography, University of Helsinki, Helsinki, Finland

*Correspondence to*: Yurui Zhang (yuruizhang@xmu.edu.cn)

**Abstract.** The Arctic stratospheric polar vortex (SPV) is a key driver of winter weather, and has been found modifying winter climate variability and its predictability in Eurasia and North America on inter-annual and decadal time scales. However, to what extent this relationship also plays a role in driving climate variability on glacial-interglacial time scales is still unknown. Here, by systematically analysing SPV changes in four sets of PMIP4 simulations for the last glacial maximum (LGM) and the pre-industrial (PI), we explore how the SPV changed during the glacial climate and how it influenced climate variability. Our results show that under LGM conditions, the SPV stretched toward the Laurentide ice sheet, accompanied by anomalous upward wave propagations and enhanced SPV variability, which increased the possibility of cold air outbreaks into mid-latitudes. During the LGM, this stretched SPV pushed cold Arctic air further equatorward, increasing winter climate variability over the more southward mid-latitudes. In particular, this strengthened winter cooling over the mid-latitudes was beyond the coverage of the Laurentide ice sheet (unlike summer). SPV-induced temperature variability also explains the inter-model spread, as removing the SPV variation from the model results reduces the inter-model spread by up to 5 °C over mid-latitude Eurasia. These results highlight the critical role of SPV in connecting the polar region and mid-latitudes on glacial-interglacial time scales. These connections are reminiscent of intra-seasonal stratosphere–troposphere coupling.

**Keywords:** polar vortex, mid-latitude climate, winter temperature, climate variability, LGM glacial climate

## 1 Introduction

The stratospheric polar vortex (SPV) is an area of high-speed, cyclonically rotating winds in both polar regions. Influenced by atmospheric waves propagated upward from the troposphere, the forming and decaying of SPV occur on the seasonal time scale (Baldwin et al., 2003; Kolstad et al., 2010; Cohen et al., 2014). In the Arctic, it has been found that the SPV forms in autumn when Arctic temperatures cool rapidly (Kolstad et al., 2010). The increased temperature difference between the polar

region and the tropics causes strong winds to develop, and the Coriolis effect causes the vortex to spin up (Baldwin et al., 2003; Baldwin and Thompson, 2009). Interacted with the troposphere, this SPV strengthens when the latitudinal temperature gradient enhances and reaches its maximum in winter (Cohen et al., 2021; Plumb, 1985; Takaya and Nakamura, 2001). The stratospheric polar vortex breaks down again in spring as the polar region warms up and the latitudinal temperature gradient decreases.

These changes in the stratospheric SPV strength can feed back to affect weather and climate closer to the Earth surface by adjusting on Arctic air intrusion into mid-latitudes (Baldwin and Dunkerton, 2001; Baldwin et al., 2003; Cohen et al., 2014). When the stratospheric vortex of the Arctic is strong, there is a single vortex with a jet stream that is well constrained near the polar front, and the Arctic air is well contained. When this northern stratospheric vortex weakens, it either breaks into two or more smaller vortices or it is displaced away from the North Pole (Cohen et al., 2021; Kretschmer et al., 2018; Cohen and

Jones, 2011). The flow of Arctic air then becomes more disorganized, and masses of Arctic air can push equatorward (Kretschmer et al., 2018; Cohen et al., 2021; Cohen et al., 2014). These seasonal dynamic changes make SPV a main driver for winter weather over mid-latitudes (Kolstad et al., 2010).

Apart from its seasonal change, the Arctic SPV is also characterized by considerable inter-seasonal and inter-annual variability (Manzini et al., 2012; Reichler et al., 2012). Influenced by the variation of atmospheric waves from the below troposphere, the

stratosphere organizes the chaotic wave forcing and creates long-lived changes (a week to several months) in the hemispheric-scale circulation (Mcintyre and Palmer, 1983; Cohen et al., 2014). This could trigger occasional breaking of stratospheric waves, analogous to ocean waves breaking on a beach, that causes stratospheric air flow becoming more disorganized, and masses of cold Arctic air can push further south, bringing with them a sharp temperature drop (Baldwin et al., 2003; Mcintyre and Palmer, 1983). These sporadic occurrences of a weak Arctic SPV event have significant impacts on surface weather and

climate variability in inter-seasonal and inter-annual time scales (Cai et al., 2024; Kolstad et al., 2010; Manzini et al., 2012). The negative phase of North the Atlantic Oscillation (NAO, defined as the surface sea-level pressure difference between the Subtropical High and the Subpolar Low) in the troposphere is found following the weakening and warming of the stratospheric polar vortex (Yang et al., 2016). In this sense, the Arctic SPV variation has been thought as an important driver of weather predictability and the climate variability on inter-annual and decadal time scale over Eurasia and North America (Kim et al.,

2014; Kim et al., 2022; Zhang et al., 2022). However, if it also plays a role in climate variability on longer timescale, such as in glacial-interglacial cycle scale with dynamic continental ice sheets, has not been explored yet.

The last glacial maximum (LGM; ~ 21 thousand years ago) is the most recent global cold extreme and has been widely documented by various proxy records (Cleator et al., 2020). The LGM world was very different from the present, with ice sheets covering northern North America and Fennoscandia, in addition to the Greenland and Antarctic ice sheets that are still

present today (Clark and Mix, 2002; Peltier et al., 2015). These extensive ice sheets represented large changes in topography and modified the spatial pattern of surface temperatures (Harrison et al., 2015; Kageyama et al., 2021). This alteration is expected to generate planetary waves that can propagate into the stratosphere and affect SPV variation, through the active interaction between troposphere and stratosphere (Baldwin et al., 2003; Baldwin and Thompson, 2009).

As for the climate of LGM, many studies, from proxy record compilations to the Paleoclimate Modelling Intercomparison Project (PMIP), and to the data-assimilation, have been carried out to investigate the spatial patterns of the cold climate features and their driving mechanisms (Harrison et al., 2015; Kageyama et al., 2021; Cleator et al., 2020; Tierney et al., 2020; Annan et al., 2022). The enhanced cooling at high-latitudes and increased land-sea temperature contrast have been identified as key features of glacial climate during the LGM, resulting from polar amplification (Kageyama et al., 2021). Another common feature emerging from proxy data and PMIP model results is the enhanced winter cooling over mid-latitudes. Proxy-based reconstructions show a 5–8 °C more temperature reduction in winter than in summer (Cleator et al., 2020). Model results from PMIP4 and data-assimilation results also reveal enhanced cooling in winter relative to summer (Annan et al., 2022; Kageyama et al., 2021; Tierney et al., 2020). The detailed structure of this enhanced winter cooling at mid-latitudes, however, has large spatial variation and varies widely among models (Annan et al., 2022; Kageyama et al., 2021; Tierney et al., 2020). For instance, two groups of data assimilation results give different LGM cooling with zonal profile differences of 3°C over the mid and high latitudes (Annan et al., 2022; Tierney et al., 2020). Therefore, the research questions to be answered in this paper are: 1) how the Polar Vortex (SPV) varied during the LGM glacial climate; 2) more importantly, how the SPV changes contributed to the climate variability during the LGM.

## 2. Methods

### 2.1 PMIP4-LGM simulations

Given its representativeness of full glacial conditions, the LGM has been the focus of the Paleoclimate Modelling Intercomparison Project (PMIP) since its inception (Braconnot et al., 2012; Braconnot and Kageyama, 2015; Harrison et al., 2015; Kageyama et al., 2017). Key climate drivers considered in these PMIP simulations include the extensive continental ice sheets, lower atmospheric greenhouse gas concentrations, and changes in orbital parameters (Kageyama et al., 2017). Compared with its precedents, the new PMIP4 experimental protocol also includes some changes in newly added forcings. For instance, PMIP4 allows vegetation and atmospheric dust loadings to change accordingly. Therefore, we focus here on PMIP4 results as these represent the most up-to-date simulations of the LGM climate.

We searched for all available PMIP4-LGM simulations from the Earth System Grid Federation (ESGF) database, and found 5 suitable simulations all together (Table S1). Given our interest in exploring the relationship between mid-latitude winter climate and polar vortices, the up-to stratospheric geopotential height and air temperature are the two most important variables. The extra variables, such as sea ice extent and sea surface pressure, can enable us to pinpoint the reasons causing these changes. With those target model output variables, five models offer downloadable data for the LGM and PI period from the ESGF database. Notably, since CESM2-FV2 and CESM2-WACCM-FV2 are from the same model family, we only include CESM2-FV2 for two reasons: 1) to maintain comparability with other PMIP models, which do not include a chemical component; and 2) a previous study found no significant climatic differences between those two version for LGM (Zhu et al., 2022). In the

end, we thus selected four simulation MIROC-ES2L (hereafter MIROC), AWI-ESM-1-1-LR (AWI-ESM), MPI-ESM1-2-LR (MPI-ESM), and CESM2-FV2 (CESM) for our analysis.

These four selected simulations run with the corresponding models that belong to a fully coupled earth system model. For the MIROC, the atmosphere module is CCSR-AGCM represented with a resolution of 128 x 64 in latitude and longitude, and with 40 vertical layers reaching a top layer of 3 hPa. The oceanic component is COCO4.9 with nested sea ice, utilizes tripolar
coordinates with 360 x 256 grids in latitude and longitude, and 63 vertical levels (Hajima et al., 2020). MPI-ESM represents the atmosphere with the ECHAM6 that has 192 x 96 grids in latitude and longitude, with 47 vertical layers reaching a top layer of 0.01 hPa. The marine module is MPIOM1.63, utilizing a grid of 256 x 220 in latitude and longitude, with 40 vertical layers (Mauritsen et al., 2019). The AWI-ESM comprises the atmospheric component ECHAM6 (same as the MPI-ESM), the ocean-sea ice component FESOM, and the terrestrial carbon model JSBACH. Both the atmospheric and oceanic components have
an average resolution of around 250 km and 100 km (Shi et al., 2020). For the CESM2-FV2, the atmospheric component is CAM6, it operates on 144 x 96 grids in latitude and longitude, with 32 vertical layers reaching 2.25 hPa. The oceanic component is POP2 and nested with sea ice module CICE5.1, features a grid of 320 x 384 in latitude and longitude and 60 vertical layers(Danabasoglu et al., 2020).

All these four simulations have been performed with the prescribed atmospheric greenhouse gases by following the protocol
given in Kageyama et al. (2017). According to ice core records, the greenhouse gas (GHG) concentrations during the LGM were overall lower than at present (Bereiter et al., 2015). The considered GHG include $CO_2$, $CH_4$ and $N_2O$, which have been prescribed as 190 ppm, 375 ppb and 200 ppb, respectively. As for the orbital parameters, all the simulations were run with the prescribed eccentricity as 0.018994, inclination as 22.949 °and perihelion as 114.42 °(Kageyama et al., 2017). These orbital configurations cause a slight decrease of summer solar radiation at Northern Hemisphere high latitudes and an increase in
winter insolation with the total magnitude of 10 W/m$^2$ (Kageyama et al., 2017). More detailed information refers to Kageyama et al. (2017; 2021).

## 2.2 Polar vortex analysis

To investigate the effect of the stratospheric polar vortex on mid-latitudinal climate, we constructed composites of the climate with weak and strong vortex activity and compare them with the average climate state. The composite procedure is based on
the vortex strength index (VSI) defined by Kolstad et al., (2010) as: VSI=$-\sum((Z-\bar{Z})cos\varphi)/\sum cos\varphi$, Z is the geopotential height, $\bar{Z}$ is its climatological mean, φ is the latitude, and the sum was performed on all grid points north of 65˚N. The reason for the minus sign is that the vortex is weak when the pressure is high and vice versa (Kolstad et al., 2010). This VSI is a conventional quantity of measuring the vortex variability and has been validated as a reliable indicator for its variation in both seasonal and inter-annual time scales (Zhang et al., 2022).

The yearly varying VSI time series data were calculated according to winter season (DJF) geopotential height at 20 hPa, since the Arctic stratospheric polar vortex shows strong seasonal variations. As shown by monthly changes VSI in Figure S1, the

VSI calculated from the PI simulations is much weaker during the summer with negative VSI and stays at a stable level of less than -1000 gpm. By contrast, it is strengthened during the winter with the positive VSI, and also shows larger inter-annual variability (Fig S1). This seasonal variability of PI simulation is similar to the results of EAR5 re-analysis data, suggesting that models be able to catch these variations. Correspondingly, we selected years of strong and weak SPV based on their winter (DJF) VSI index using the one standard deviation ($\sigma$). The strong SPV years are represented by those VSI larger than $\sigma$, while the weak SPV years are given by those below $\sigma$ (indicated as red and green dots in Fig S2). It is worth to notice that MIROC shows much less inter-annual variability than the other models. For the PI simulation, MIROC has the $\sigma$ of 70 gpm, which is only 1/3 of as in other models. Further comparison with ERA5 re-analysis suggest that MIROC seems to underestimate their inter-annual variabilities (Fig S2). Similar to the previous analysis, we composite all the weak and strong SPV years to denote the climate state of weak and strong vortex, respectively (Kolstad et al., 2010; Zhang et al., 2022).

## 3. Results and Discussions

### 3.1 Arctic polar vortex stretched over ice sheets during the LGM

As represented by troughs of low geopotential height of the stratosphere (e.g. at 20 hPa), the winter polar vortex was sitting over the Arctic in the PI climate. The smallest geopotential height is around 245 gpm, with inter-model ranging from 244 gpm in MIROC to 246 gpm in AWI-ESM. Triggered by the asymmetric troposphere-stratosphere wave flux between Asia and North America, the SPV is not a perfect circular but instead stretched toward North America (Fig. 1a). An upward wave energy over Asia that is reflected downward over North America drags the center of SPV slightly shifts toward the Atlantic side (Jones and Cohen, 2011; Kretschmer et al., 2018). This overall pattern fits the ERA5 re-analysis data, as shown by the similar shape of 250 gpm contour (black line in Fig. 1a). The slightly more extensive SPV during the PI climate compared to the more recent climate used for the re-analysis is expected, given that the PI climate was a bit colder than the climate for 1940-2024 used for ERA5. This is consistent with the results of 3 out of 4 models (CESM being the exception) (Fig. 1).

Compared to PI, one significant feature of SPV during the LGM is that it was stretched toward the American continent. For instance, the contour line of 250 gpm was stretched toward N America by 4-8 °latitude. Accordingly, the center of SPV shifts toward America. This ice-sheet-related stretching of SPV seems increase temporal VSI variability, as indicated by a larger standard deviation in three LGM simulations compared to in their PI simulations. The exception is CESM, which exhibits greater temporal variability during the PI period (much larger than observation and other models) than LGM (Fig. S2). Compared among different models, the magnitudes of their SPV responses during the LGM are consistent with the standard deviations of their inter-annual variabilities (Fig. S2). For instance, the relatively small LGM response of SPV in MIROC is in line with its smaller inter-annual variability with standard deviation of 50 gpm.

It has been illustrated that polar stratospheric variability is largely dominated by vertically propagating Rossby waves of tropospheric origin (Charney and Drazin, 1961; Cohen et al., 2007). During the LGM, the presence of the Laurentide ice sheet

in the troposphere, up to 2-3km height, could generate planetary waves in the troposphere by modifying the topography (Mcintyre and Palmer, 1983; Polvani and Waugh, 2004). Therefore, the ice-sheets-related planetary wave changes could affect the SPV, which can in turn descend and influence the surface climate.

The stretched SPV during the LGM climate seems compatible with other studies on polar vortex variations relevant to the climate states. For instance, previous studies on SPV decadal variability have shown that the Arctic polar vortex shifted towards the Eurasian continent and away from North America in response to the Arctic warming and sea-ice loss, particularly over the Barents–Kara seas in recent decades (Zhang et al., 2016a). Our study reveals that the SPV shrunk during the deglaciation from the LGM to the present, suggesting that the SPV evolved in the same direction as it evolved over the past few decades, despite different mechanisms (Manzini et al., 2012). Therefore, together with previous relevant studies, our results suggest that a warmer climate favors SPV shrinkage and shift towards Eurasia.

Previous studies have demonstrated that the vertical wave activity flux (WAF) is the key determining the strength of the troposphere-stratosphere interaction (Baldwin et al., 2003; Jones and Cohen, 2011; Polvani and Waugh, 2004). To investigate the wave energy interface between the troposphere and stratosphere, we calculated December Plumb WAF anomalies at 100 hPa by following Plumb (1985). Plumb WAF results of the PI simulation show an overall positive vertical WAF over N Atlantic and Eurasia, indicating upward propagation from Rossby waves. The downward WAF over the Pacific and North America denotes downward reflected planetary waves there (Fig. 1b). This simulated Eurasia-America asymmetric pattern in the PI climate state is roughly consistent with ERA5 re-analysis data with a similar location of zero-contours (Fig S3). The main difference is that the ERA5 re-analysis shows a larger magnitude for both positive and negative WAF than PI simulations, and this underestimation is a common issue for coupled climate models. This simulated Eurasia-America asymmetric spatial pattern fits our current understanding on troposphere-stratosphere coupling (Cohen, 2007; Jones and Cohen, 2011; Kretschmer et al., 2018). Compared with the PI simulation, the vertical WAF in the LGM simulation is enhanced over the Eurasian continent in all model results, implying more wave energy propagates upward to the stratosphere. Meanwhile, vertical WAF during the LGM is reduced over the Pacific Ocean, indicating less wave energy propagates there. This wave energy propagating into the stratosphere is supposed to lead to anomalous warming and weakening of the SPV (Polvani and Waugh, 2004). This seems unusual given the overall cooler background climate, as previous studies have suggested that a warming climate could weaken SPV strength by enhancing turbulent heat flux (e.g. Kim et al. 2014; Kug et al., 2015). If a warming climate weakens the SPV, we would expect it to strengthen in a cooler climate. However, we found that the anomalous upward WAF response during LGM occurred likely due to the existence of the ice sheets on the continents, which altered the topography and surface properties, potentially outweighing the effect of background climate. Previous studies have found that alternation of topography and snow cover could induce anomalous WAF and SPV responses (Cohen, 2007; Allen and Zender, 2010; Cohen and Jones, 2011; White et al., 2018; Pan and Duan, 2023).

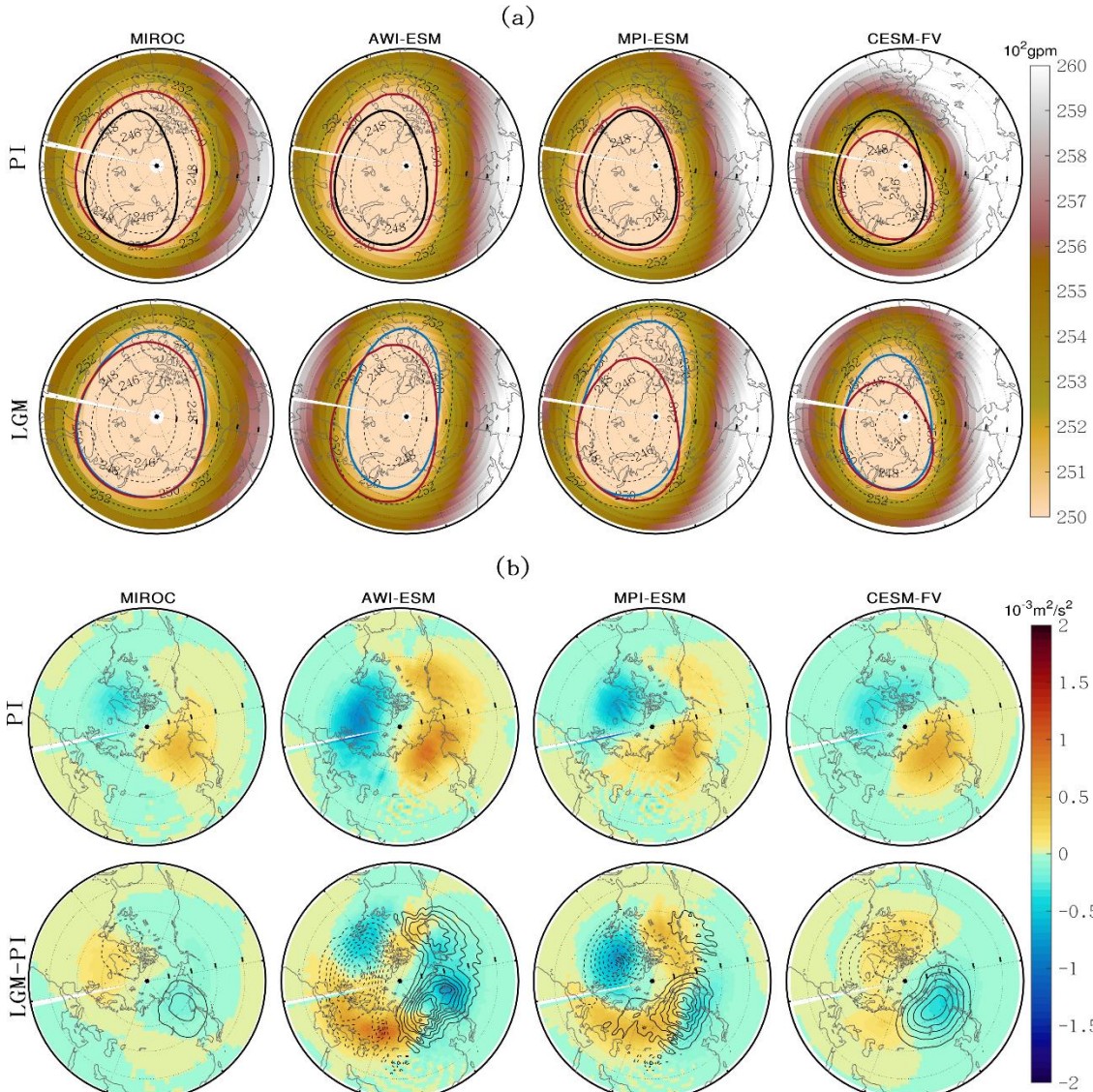

**Figure 1** a) Geopotential height at 20hPa in the PI (upper panel) and LGM (lower panel) simulations, illustrating the shape and strength of the Polar Vortex. The dashed contour lines denote $2*10^2$gpm intervals from 246 to $252*10^2$gpm. Red and blue lines (in the lower panel) refer to $250 *10^2$gpm for PI and LGM, respectively. The black line in the upper panel refers to the ERA5 re-analysis data for the period of 1940-2024. b) The upper panel shows the December Plumb wave activity flux anomalies (WAF, in 1e-3 m$^2$/s$^2$) at 100 hPa in the PI simulations, with positive values indicating upward wave energy flux that vertically propagates from Rossby waves, and negative values giving downward reflected planetary waves. The lower panel presents their LGM WAF anomalies from PI (LGM-PI). The black and dashed contours represent positive and negative WAF at intervals of 0.1 1e-3 m$^2$/s$^2$ from the PI simulations for reference.

### 3.2 The impact of polar vortex on mid-latitude climates

### 3.2.1 Stretched SPV enhanced the LGM winter cooling

Compared to the PI, the LGM climate was significantly cooled by the ice sheets, lower GHG, elevated atmospheric dust, and related feedback processes between different components of the climate system. The simulated temperature was cooled by more than 15℃ over the ice sheets. However, a closer look at the temperature changes in the simulations and in proxy-based reconstructions shows a stronger LGM cooling in winter than in summer (Fig. 2). The LGM summer cooling was strictly constrained over the ice sheets, highlighting the controlling effect of ice sheets on summer climate. The primary mechanism of the ice sheets cooling the climate includes elevated altitudes, an enhanced ice-albedo feedback and modified atmospheric circulation (Renssen et al., 2009; Zhang et al., 2016b). By contrast, this enhanced cooling extended further into the mid-latitudes of the continents during the winter. For instance, the -10℃ isotherm line of the LGM winter temperature anomalies extends down south to nearly 30 °N in North America that is far beyond the coverage of the Laurentide ice sheet (Fig. 2). This implies that extra processes play a role in cooling the mid-latitudes during winter in addition to the direct ice sheet cooling effects. From the climate forcing perspective, orbital scale insolation could potentially induce seasonal and latitudinal change up to this magnitude. Nevertheless, a very similar LGM orbital setting to the present day (as discussed in Section 2.1) excludes this possibility.

The enhanced LGM cooling over mid-latitudes is probably linked to the stretched polar vortex that is an important driver for mid-latitude climate. First, planetary waves generated by the presence of continental ice sheets could extend the southern boundary of regions where cool air could arrive further south and induce cooling at mid-latitudes (Kolstad et al., 2010). Furthermore, the ellipse shape of SPV stretched by the ice sheet can create irregular waves when propagating that is less stable than the round shape SPV of the present (Zhang et al., 2022). Both of those factors can lead to an enhanced winter cooling at the mid latitudes during the LGM. The SPV-related large variations of winter cooling among these four models contribute to the inter-model spread (will be discussed in Section 3.3.2).

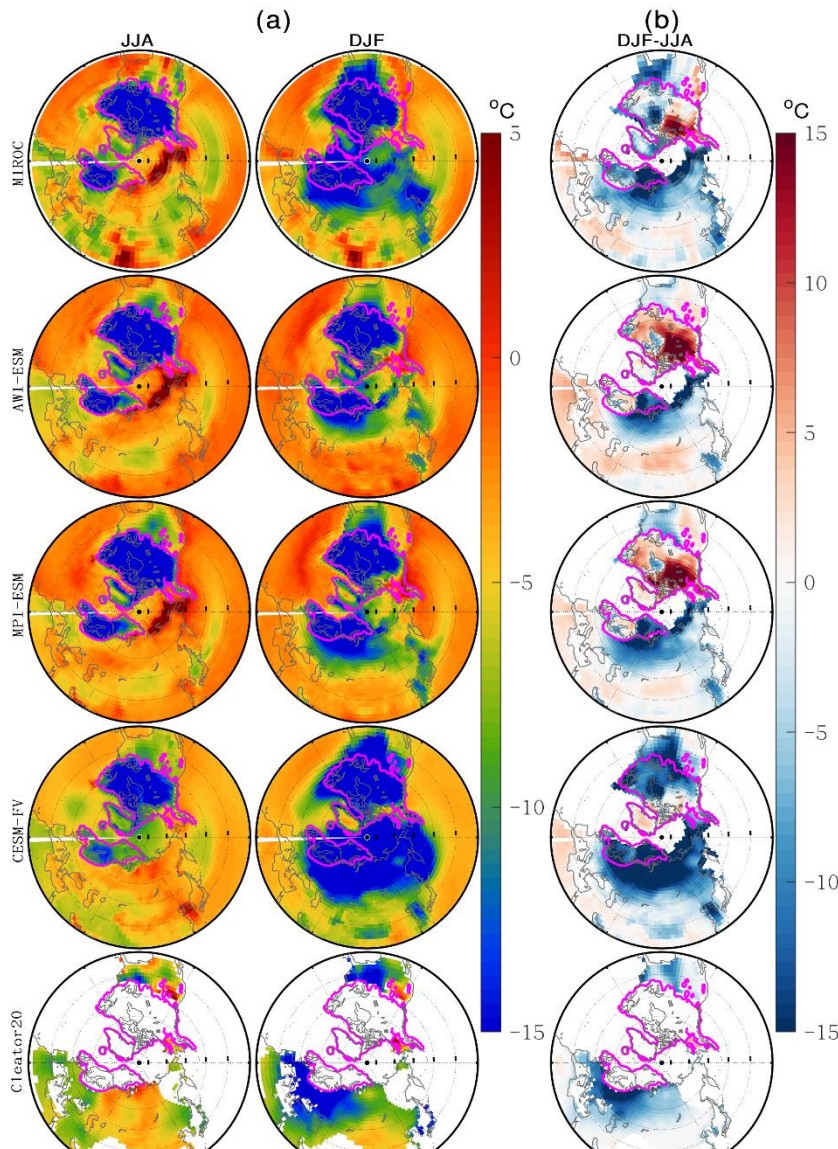

**Figure 2** Seasonal LGM temperature anomalies (LGM-PI) (a) and their seasonal differences (b, defined as DJF-JJA from (a)). The upper panels show results from the simulation and the last panel presents the data-assimilation of Cleator et al. (2020). The magenta contour line marks the range of ice sheets during LGM.

### 3.2.2 Linkage of polar vortex variation to winter climate

The composite analysis (i.e. composited winter temperature anomalies) reveals that the effect of a weak SPV on climate is shown as dipole pattern between mid- and high latitudes for the Eurasian and N American continents, respectively (Fig. S3). This effect can be further illustrated by the surface air temperature (SAT) difference between the weak and strong SPV

composite (Fig. 3). The weak SPV causes positive temperature differences of ~ 1°C in the south and negative differences up to 2-3°C in the north with the boundary at 40-45 °N over Eurasia. In N America, weak SPV causes warm conditions over high latitudes and cold conditions in lower latitudes with a boundary ranging from 45 °N in MPI to 30 °N in MIROC.

These patterns can be explained by the atmospheric circulation, as indicated by sea level pressure changes (Fig. 3). A weak SPV can result in a weakening of the subpolar low by inducing positive anomalies in sea level pressure and an increase of the subtropical high by prompting negative anomalies over the North Atlantic. This increased polar high facilitates the flow of colder air from the Arctic into Eurasia, while a decreased subpolar low can reduce heat carried by the air from the south. These two processes together cause dipole responses on the Eurasian continent. For the North American continent, the upstream
region of the Pacific shows opposite responses and the temperature shows a contrasting dipole pattern (Fig. 3). The results of the winter temperature and surface circulation responses to SPV variation fit observational analyses on inter-season and inter-annual variations. The observational analyses have shown a strong connection between the strength of the stratosphere polar vortex and the dominant pattern of surface weather variability, such as the North Atlantic Oscillation (NAO) (Yang et al., 2016). In this sense, the slowly varying stratospheric signal may help predict the North Atlantic Oscillation changes and the
weather for the coming months (Baldwin and Dunkerton, 2001).

During the LGM, the overall pattern of composite temperature anomalies for weak and strong PSI is similar to the present day (Fig. 3). However, the strength of temperature anomalies is generally enhanced over Eurasia in most of the models, with the exception of CESM. For instance, the cooling anomalies are increased from 2°C to 3°C in MIROC. Some other visible differences include the spatial range of warming and cooling. For instance, the negative temperature anomalies over Eurasia
extend slightly further south in the LGM than in the PI. Therefore, the difference in surface winter temperature between strong and weak SPV conditions during the LGM is up to 2-4°C in general, which is slightly stronger than PI.

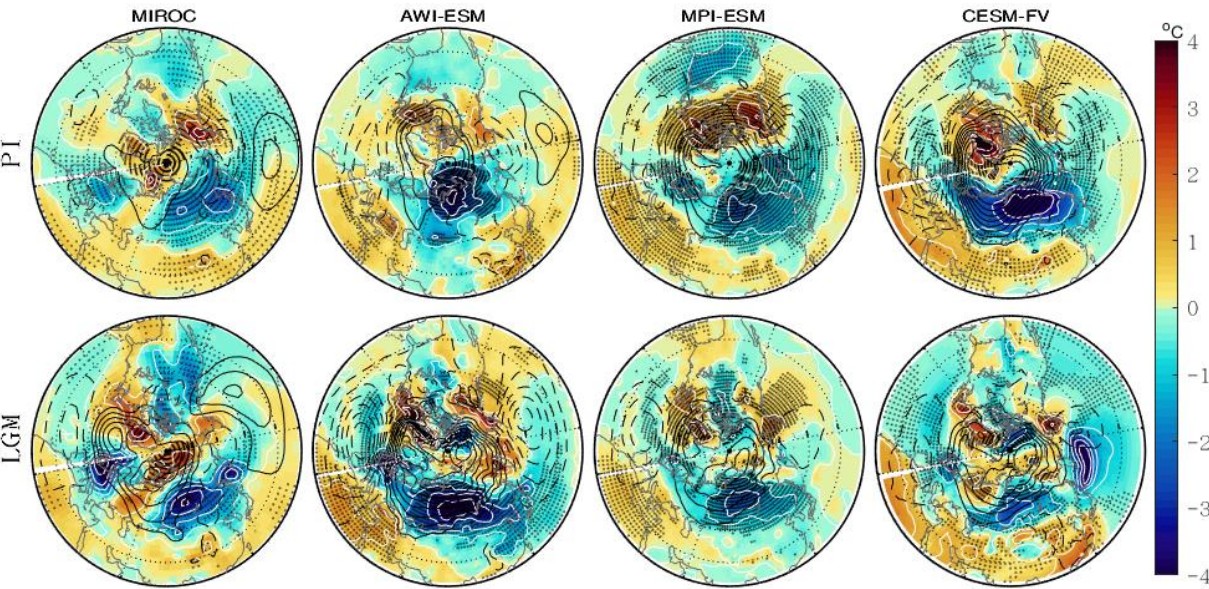

**Figure 3** Winter surface air temperature (SAT) differences (filled colour, in °C) between the weak (low PSI) and strong (high PSI) polar vortex composites for both PI and LGM with significance denoted by dots. The sea level pressure (SLP) (contours, with the interval of 120 hPa) in DJF is also shown.

### 3.3 SPV variation increases climate variability and uncertainty

### 3.3.1 Large inter-annual climate variability at mid-latitudes

The spatial distribution of the temperature standard deviation shows large climate variations over mid-latitudinal continents and near the margin of sea ice extension (Fig. 4a). Compared with PI, climate variation during the LGM is overall enhanced over both land and ocean at mid-latitudes. The region of large climate variability moves southward during LGM. This leads to dipole patterns of the temperature variability differences between LGM and PI, with significant enhancement in the south and a reduction in the north (Fig. 4b). The enhanced climate variation over the North Atlantic during the LGM is in line with the
southward extended sea ice margin (Fig. 4a & 4b). This implies a controlling role played by dynamic sea ice in climate variability.

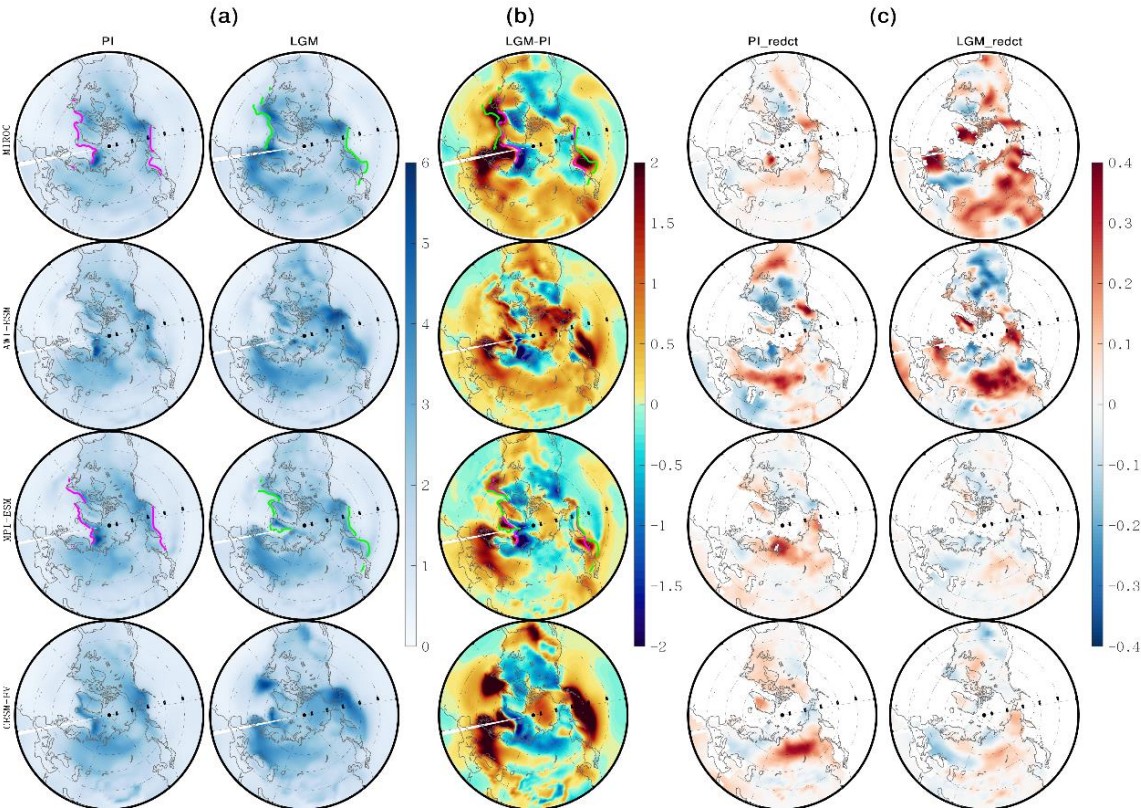

**Figure 4** Winter month climate variability, shown as Standard Deviation (SD) of winter temperature for PI and LGM **(a)**, their LGM anomalies **(b**, calculated by (a)-(b)), and the SD reduction **(c)**. The SD reduction is defined as the SD difference between the total standard deviation and those of the strong and weak SPV years (i.e. used for the composites) are removed. Magenta and green lines in (a) and (b) indicate the margin of sea ice during winter for the simulations when available (MIROC and MPI-ESM in this case).

On the land, the enhancement of the LGM climate variability at mid-latitude of Eurasia and North America is visible and probably related to the stretched SPV (Fig. 4a & 4b). Removing the strong and weak SPV composites can clearly reduce the climate variability over the mid-latitude continents (Fig. 4c). The common feature of this reduction among models is their positive anomalies, despite some variations in their detailed pattern. Compared with PI, the region with the large reduction is located further south during the LGM, which is consistent with a stretched SPV. This enhanced climate variability during the winter over the mid-latitudes also appears in previous studies on seasonal climate of LGM (Cleator et al., 2020; Kageyama et al., 2021). The winter temperature variability has been found to be enhanced over the mid- and high-latitudes, which further contributes to a large part of LGM climate variability (Annan et al., 2022; Cleator et al., 2020; Kageyama et al., 2021).

### 3.3.2 Large inter-model spread of winter temperature

Although the four models give similar patterns of LGM temperature anomalies in general, there are still visible differences. The root mean square deviation of individual models' temperature shows that the winter has a larger inter-model spread than the summer (Fig. 5). The largest inter-model spread was found over the Nordic Seas, which is due to large climate uncertainty induced by dynamic sea ice. Over the continents, the inter-model spread is overall larger than over the oceans, and large values are found over mid-latitudes. This wide spread over mid-latitudes is the primary contributor to climate uncertainty.

The large spread of multi-model temperatures over the mid-latitudinal continents can be significantly reduced by removing the strong and weak SPV composites (Fig. 5b). For zonal mean temperature, the winter inter-model spread decreased by almost 0.8°C for the latitude of 30-50 °N when removing the weak and strong SPV years. In particular, the reduction over North America is up to 5 °C, and mainly distributed over 30-45 °N (Fig. 5b). For the Eurasian continent, a similar degree reduction is found over the latitude of 40-65 °N.

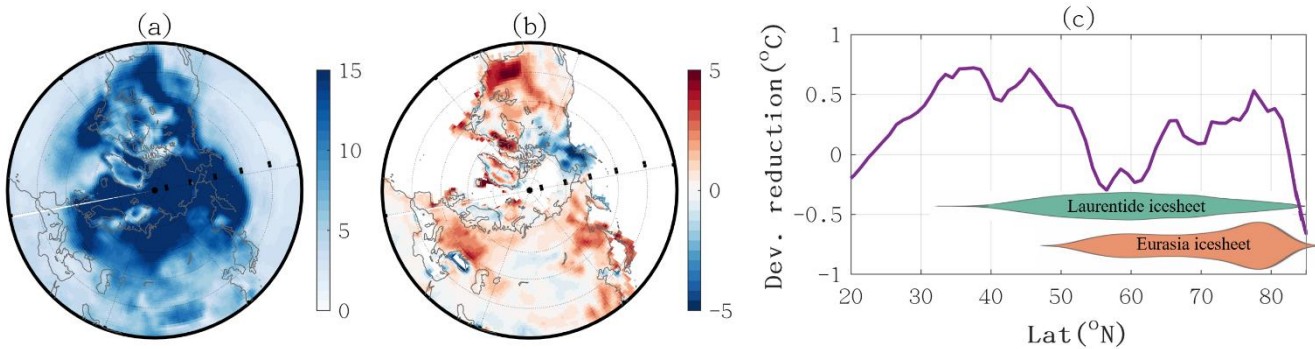

**Figure 5** Inter-model spread of LGM temperature anomalies for winter (DJF). **(a)** is the inter-model spread of LGM temperature anomalies, defined as the root mean square deviation of LGM anomalies from ensemble mean. **(b)** is the difference between the result in (a) and inter-model spread after removing their weak and strong VSI composites. **(c)** is the latitudinal profile of (b) over the continents, overlaid with the ice sheet latitudinal position during LGM.

Previous studies have found a less consistent winter climate among different models than summer (Harrison et al., 2015; Kageyama et al., 2021). Our analyses identify a previously unknown source for mid-latitudinal climate variability and inter-model spread, suggesting that SPV variations need to be taken into account for realistic simulation of climate variability during the glacial-interglacial cycle.

### Conclusions

Our analysis of the polar vortex (SPV) changes during the LGM suggest that it has a key role in linking the polar and mid-latitudes even in glacial climate and its adaptation according to the glacial-interglacial cycle. SPV weakening causes two dipole

structures of continental winter temperature for the LGM that are similar to PI, suggesting that the mechanisms of SPV affecting the tropospheric climate works for both periods. Comparison of SPV between the PI and LGM shows that the during the LGM SPV was stretched toward the American continent, which pushes the regions affected by outbreaks of cold air further south. The difference in surface winter temperature between strong and weak SPV conditions during the LGM is up to 2-4°C,

which is slightly stronger than during the PI. During the LGM, the stretched SPV pushed cold Arctic air further equatorward, increasing the mid-latitudinal winter climate variability. Removing the SPV variations can reduce inter-annual variability of winter temperature up to 5°C over mid-latitude Eurasia. SPV-induced temperature variability also explains the inter-model spread, as removing SPV variation persistently reduces the winter temperature variation (root mean square deviation) mid-latitudes by 0.8°C. These results highlight the critical role of SPV in linking the polar and mid-latitudes even in glacial-

interglacial timescale.

## Code and data availability

PMIP4 simulation data are available on the Earth System Grid Federation (ESGF) website: https://esgf-node.llnl.gov/projects/esgf-llnl/. The LGM assimilation data used in this study are from Cleator et al. (2020) and can be accessed at https://doi.org/10.17864/1947.244.

## Author contributions

YZ, HR and HS designed the study. YZ conducted the analysis and wrote the first draft of the paper. ZL and XL proceeding model data under YZ's guidance. All authors contributed to interpreting the results and writing the manuscript.

## Competing interests

The contact author has declared that none of the authors has any competing interests.

## Acknowledgements

The authors would like to thank Judah Cohen, Yong Song, anonymous referees and the editor Qiong Zhang for their valuable comments that have greatly improved the manuscript. The authors acknowledge PMIP4 and specifically to the involved modelling groups for running the experiments and for making their results available for further analysis. This study was funded by the National Key Research and Development Program of China (2023YFF0804600) and MEL Internal Program

(MELRI2403).

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
