# Peer review of "Stretched polar vortex increases mid-latitude climate variability during the Last Glacial Maximum"

_Climate of the Past, 2024_

## Referee Comment (RC2)

**General comments**

The observation data and numerical modelling have suggested that the Arctic stratospheric polar vortex (SPV) is playing role in inter-seasonal variability and predictability of the winter climate over Eurasia and North America. By analysing PV changes in the PMIP paleo-simulations, the author explored the PV changes and its influences on climate variability during the glacial climate. The results show that under LGM conditions, the PV stretched toward the Laurentide ice sheet increased the possibility of cold air outbreaks into mid-latitudes. This finding provides an explanation to the observed extreme winter cooling and long-stand inter-model spreads. The paper is well-written. I recommended to published it with the following minor revision.

**Comments to authors:**

Line 10: The abbreviation of polar vortex PV could be confused with PV of potential vorticity, suggested change to SPV.

Line 18: should be "was beyond…"

Line 86: Table 1 was not included. Should be Table S1?

Line 102: Present AWI-ESM resolution in the form of grid numbers, like for the other models.

Lines 122-132: add unit of gpm to VSI? like -1000 gpm and 70 gpm

Line140: Further specify that ERA5 is shown in black line. For instance, "This overall pattern fits the ERA5 re-analysis data, as shown by the similar shape of 250 gpm contour (black line in Fig. 1).

Line 172-174: "…previous climate models results found that the stratospheric polar vortex itself can be either colder or stronger with increasing GHG depends on the strengthen amplitude of the troposphere originated planetary waves (Baldwin et al., 2003). The expression is not very clear, please clarify.

Line 211:"warm-cold-warm-cold pattern" sound weird, do you mean dipole between mid- and high latitudes?

Figure 3: add confident level to the figure.

Add a figure to illustrate how PV different between the LGM and PI affect the climate. Calculating wave activity fluxes by focusing on the North American to Eurasian cross section to analyze stratosphere-troposphere interactions?

---

## Author Comment (AC1)

The manuscript presents an analysis of archived model data that participated in the PMIP project. The authors propose the Laurentide ice sheet resulted in more stretching of the polar vortex (PV), which contributed to colder and more variable temperatures in eastern North America south of the ice sheet.

It is my impression that the authors are not that familiar enough with stratosphere-troposphere coupling and polar vortex variability to adequately interpret their model analysis of stratosphere-troposphere coupling. Scientists well versed on stratosphere-troposphere coupling would not make the omission mentioned in my first minor comment. A stretched polar vortex was first described in Cohen et al. 2021 and that paper and Kretschmer et al. 2018, at a minimum need to be read carefully and cited. But there are many other recent papers that should be included.

We thank the reviewer for these comments and suggested citations. We were actually focused on the temporal dynamics such as seasonal developing and waning processes as that will directly relate to the strengthening and weakening variation of PV, and did not discuss different types of PV weakening. But we agree that adding these citations will make the study more comprehensive. We have thus added these citations (attached at the end of this document) and included a discussion on two types of PV weakening.

I aso think the authors chose poorly when they analyzed CESM-FV2 and ignored CESM2-WACCM-FV2. WACCM was designed with improved simulation of troposphere-stratosphere coupling. Whether the goal was achieved is still an open question but if at all possible the authors should include analysis of CESM2-WACCM-FV2 in their study.

We did not ignore the CESM2-WACCM-FV2 version. On the contrary, we were indeed aware of these two model versions and investigated their relationship (similarities and differences), as we stated in our manuscript: "Notably, since CESM2-FV2 and CESM2-WACCM-FV2 are from the same model family, we only include CESM2-FV2 because the other version included a module for stratospheric chemistry not included in the other PMIP4 simulations (Zhu et al., 2022).". We finally chose the CESM2-FV2 version based on two reasons: 1) we want keep it comparable with other PMIP models, as other models do not have the chemical component of CESM2-WACCM-FV2; 2) the previous study has specifically compared those two model version results for LGM and did not find significant climatic difference between them (Zhu et al., 2022). We agree that CESM2-WACCM-FV2 has the higher model top and superior stratospheric representation, but the main purpose of this detailed presentation is to include atmospheric chemical mechanisms of varying complexity and a prognostic stratospheric aerosol capability, as presented in their official webpage and the literature (Danabasoglu et al., 2020; Gettelman et al., 2019). In other words, their detailed stratospheric representation is specifically for better representing

chemical components. Anyway, we fear that the CESM2-WACCM-FV2 version would require a separate discussion on atmospheric chemical processes and their aerosol feedback, which is still an open question, as the reviewer pointed out. Therefore, we think that using CESM-FV2 is justified,

I found the paragraph describing polar vortex variability during Arctic amplification, LGM and PI starting on line 161 as conflating of different ideas.   The authors state contradicting ideas that the polar vortex strengthens both in LGM when the climate was colder and during the present period of Arctic warming when the climate is warmer.

We mean to say that a warming climate accompanied by reduced sea ice [SHT2] (e.g. current global warming) leads to weakened variability, while the cooling in the LGM seems to favor large variability indicated by large standard deviation, but not the strengthening or weakening of PV itself. And the following are a few examples from literature illustrating how warming could affect PV variation. We have rephrased this to avoid confusion.

Also the ideas presented in Thompson et al. 2000, I would argue have not aged well with time.   They argued that increasing GHGs would strengthen annular modes in both the troposphere and stratosphere, neither of which has been observed in the quarter of a century since that work was published.   Also strengthening planetary waves should weaken the polar vortex not strengthen it.

Thanks for the point on the ideas of Thompson et al. (2000). But here, we only cite their results on see-saw changes between the Arctic and the surrounding zonal rings. Nevertheless, the effect of increased GHG has been separately mentioned late in this paragraph of the previous version. "…previous climate models have found that the stratospheric polar vortex itself can be colder or stronger with increasing GHG depends on the strengthen degree of troposphere originated planetary waves (Baldwin et al., 2003)". As for the relationship between planetary waves and polar strength, that is right, strengthening planetary waves will weaken the polar vortex, but here we highlight that the existence of ice sheets perturbs the wave activity flux, as shown by our newly added Fig. 1b and thus affects their variability. We have revised this section accordingly.

The paragraph starting on line 197 describes the Laurentide ice sheets amplifying wave energy that leads to PV stretching. PV stretching over North Ameirca is mainly trigerred by upward wave energy over Asia that is reflected downward over North America not by local upward wave energy. So what the authors are presenting is a novel idea of PV stretching that has no support from the scientific literature.   In a future version of the manuscript the authors need to show three dimensional wave activity flux in the LGM experiments to understand changes in wave reflection and stretched PVs in the LGM compared to PI or even in the ERA5 reanalysis.

This is a great point. We have added the wave activity flux figure as Fig. 1b. We can see an enhanced upward WAF over W Eurasian and reduced downward WAF over North America during LGM, indicating that the continental ice sheets influence the troposphere-stratosphere interaction. With this anomalous WAF, SPV is supposed to be weakened, which is further consistent with a more dynamic mid-latitude climate. This further implies that the climate system is still behaving in the same way as present-day, but the existence of ice sheets alternative their topography and its surface properties.

Colder temperatures in the atmospheric column will lower geopotential heights in the stratosphere.   Therefore much colder tropospheric air temperatures over the Laurentide ice sheet can lower geopotential heights in the polar stratosphere in the same region giving the appearance of a stretched PV over North America. However the cold temperatures over North America are not related to or caused by PV variability.   This is not what the authors in the manuscript are discussing but this scenario needs to be excluded for the reason for colder temperatures over North America in the analysis.

The ice sheet itself definitely can lower temperature by thermodynamics such as enhanced albedo and adjusting atmospheric circulation, which has been extensively discussed in earlier studies, as cited in the manuscript. Our argument is that LGM winter temperature cooling was even more than in summer, i.e. asymmetric cold temperature between summer and winter. We have rephrased the sentence to avoid this confusion.

I have some more minor suggested edits below.   I cannot recommend that the manuscript be published in its current form.   The authors need to familiarize themselves better with the scientific literature on PV variability in general and stretched PVs in particular.   With a better understanding of stratosphere-troposphere coupling, their interpretation of the analysis will be consistent with up to date thinking on PV variability.

Those minor comments have been fixed. We also

Minor comments:

1. Line 37 – there are two types of polar vortex (PV) disruptions or weakenings, one where it splits as described in this sentence but also where the PV can be displaced away from the North Pole while remaining intact or in one piece.

   Thanks for this comment. We have added the second type of PV weakening into the manuscript.

2. Line 66 – what is actually meant by "polar amplification?" Currently the term is most commonly used to describe amplified polar or Arctic warming which is

contradicted by the beginning of the sentence that states "cooling at high latitudes."

As explained above, the polar amplification basically means that temperature change in polar regions is more expressed than at low latitudes in general, which implies these changes can be either warming or cooling, especially for the paleoclimate perspective. Under the current global warming trend, it means polar regions are warming more, but when considering a cooling trend, it means polar regions cooling more than other regions. It is clear that during the LGM the temperature change is in the opposite direction compared to the current climate changes. We have further clarified this in the revised version.

3. Table 1 – I couldn't find Table 1? Did the authors mean Table S1?

   Apologies for this bug. Table 1 has been decided to move to SI, so it should be Table S1. We have fixed this.

4. Lines 90-91 – it is my understanding that WACCM has a better resolved stratosphere than other versions of CESM2 and it is not just a chemistry model. On the NCAR web page it states that CESM2-WACCM-FV2 has 70 vertical layers and not 32 levels as in CESM2-FV2 with a much higher lid than 2.25 hPa as in CESM2-FV2.   If I am correct then the authors should not have discarded WACMM in their analysis.

   As we explained above, we know that CESM2-WACCM-FV2 has more vertical layers and a more complicated resolved stratosphere than CESM2-FV2. But here the top level of 2.25 hPa in CESM-FV2 is comparable with 3hPa of other models. In addition, a previous study by Zhu et al. (2019) has shown that further detailed representation results in no clear climatic differences. Therefore, we think our option with CESM2-FV2 is a fair choice.

5. Figure 1 – I really had a hard time reading this figure. I had to greatly blow up the figure to see the contour lines. Also the caption says that the red, black and blue lines represent the ERA5, yet they vary from plot to plot, how is that?

   We have made the contour line bold and enlarged the figure for better visualization.

   Regarding the caption, it said "Black, red and blue lines refer to 250 *10e2gpm for ERA5 re-analysis data (for the period of 1940-2024), PI and LGM, respectively." This means three colors corresponding to the three dataset of ERA5, PI and LGM. And the same colors indeed stay the same for each model. We have clarified this by adding the corresponding notes in the text, such as: "This overall pattern fits the ERA5 re-analysis data, as shown by the similar shape of 250 gpm contour (black line in Fig. 1)".

References:

Kretschmer, J. Cohen, V. Matthias, J. Runge and D. Coumou, 2018b: The different stratospheric influences on cold extremes in northern Eurasia and North America, *npj Climate and Atmospheric Science*, doi: 10.1038/s41612-018-0054-4.

Cohen, J., L. Agel, M. Barlow, C. I. Garfinkel, I. White. 2021: Linking Arctic variability and change with extreme winter weather in the US, *Science*, **373** (6559), 1116–1121, DOI: 10.1126/science.abi9167.

These literature references have been added.

Newly added refs:

Allen, R. J. and Zender, C. S.: Effects of continental-scale snow albedo anomalies on the wintertime Arctic oscillation, Journal of Geophysical Research: Atmospheres, 115, 2010.

Cohen, J. and Jones, J.: Tropospheric Precursors and Stratospheric Warmings, Journal of Climate, 24, 6562-6572, 2011.

Cohen, J., Agel, L., Barlow, M., Garfinkel, C. I., and White, I.: Linking Arctic variability and change with extreme winter weather in the United States, Science, 373, 1116-1121, 2021.

Cohen, J., Screen, J. A., Furtado, J. C., Barlow, M., Whittleston, D., Coumou, D., Francis, J., Dethloff, K., Entekhabi, D., Overland, J., and Jones, J.: Recent Arctic amplification and extreme mid-latitude weather, Nature Geoscience, 7, 627-637, 2014.

Kretschmer, M., Cohen, J., Matthias, V., Runge, J., and Coumou, D.: The different stratospheric influence on cold-extremes in Eurasia and North America, npj Climate and Atmospheric Science, 1, 2018.

Pan, Z. and Duan, A.: Influence of the Tibetan Plateau on the coupling of the North Pacific–North Atlantic pressure systems, Atmospheric Research, 295, 2023.

Polvani, L. M. and Waugh, D. W.: Upward Wave Activity Flux as a Precursor to Extreme Stratospheric Events and Subsequent Anomalous Surface Weather Regimes, Journal of Climate, 17, 3548-3554, 2004.

White, R. H., Battisti, D. S., and Sheshadri, A.: Orography and the Boreal Winter Stratosphere: The Importance of the Mongolian Mountains, Geophysical Research Letters, 45, 2088-2096, 2018.

---

## Author Comment (AC2)

**Review of the manuscript entitled "Stretched polar vortex increases mid-latitude climate variability during the Last Glacial Maximum" by Zhang et al.**

**Summary:**

The observation data and numerical modelling have suggested that the Arctic stratospheric polar vortex (SPV) is playing role in inter-seasonal variability and predictability of the winter climate over Eurasia and North America. By analysing PV changes in the PMIP paleo-simulations, the author explored the PV changes and its influences on climate variability during the glacial climate. The results show that under LGM conditions, the PV stretched toward the Laurentide ice sheet increased the possibility of cold air outbreaks into mid-latitudes. This finding provides an explanation to the observed extreme winter cooling and long-stand inter-model spreads. The paper is well-written. I recommended to published it with the following minor revision.

**Comments:**

Line 10: The abbreviation of polar vortex PV could be confused with PV of potential vorticity, suggested change to SPV.
Done

Line 18: should be "was beyond…"
Done

Line 86: Table 1 was not included. Should be Table S1?
Yes, it is supposed to be Table S1, which has been fixed.

Line 102: Present AWI-ESM resolution in the form of grid numbers, like for the other models.

done

Lines 122-132: add unit of gpm to VSI? like -1000 gpm and 70 gpm
done

Line140: Further specify that ERA5 is shown in black line. For instance, "This overall pattern fits the ERA5 re-analysis data, as shown by the similar shape of 250 gpm contour (black line in Fig. 1).
Thanks for this comment on improving the clarity. We have included this in the revised version.

Line 172-174: "…previous climate models results found that the stratospheric polar vortex itself can be either colder or stronger with increasing GHG depends on the strengthen amplitude of the troposphere originated planetary waves (Baldwin et al., 2003). The expression is not very clear, please clarify.

We mean that the wave activity flux plays a key role in determining the SPV strength. In this revised version, we decided to stay away from GHG increase case discussion as that can be a quite complicated issue. So this sentence has been replaced by "Previous studies have demonstrated that wave activity flux is the key to determine strength of troposphere-stratosphere interaction (Baldwin et al., 2003; Jones and Cohen, 2011; Polvani and Waugh, 2004)."

Line 211:"warm-cold-warm-cold pattern" sound weird, do you mean dipole between mid- and high latitudes?
yes, that is exactly what we mean. We have rephrased the sentence by following this comment.

Figure 3: add confident level to the figure.
Done for the figure 3.

---

## Author Response (AR2)

Review#2

The manuscript presents an analysis of archived model data that participated in the PMIP project. The authors propose the Laurentide ice sheet resulted in more stretching of the polar vortex (PV), which contributed to colder and more varaible temperatures in eastern North America south of the ice sheet.

I found that the revised manuscript is greatly improved from the original submission. I would just advise the authors that the WAF looks a bit strange to me from the different models as shown in Figure 1. This doesn't surprise me as models struggle with this. Kretschmer et al. 2018 show the WAFz at 100 hPa in Figure 2 for stretched SPV events. I was looking for a good figure showing the climatology but wasn't successful. In general it is up over northern Asia and downward over northern North America. See also Figure 1 from Cohen et al. (2007). I just ask that the authors be mindful of the differences from what they show in the PMIP models and what the observations show. A supplementary figure on the difference between the model climatology and that derived from ERA5 might provide some useful context to the reader.

We thank the reviewer for this greatly relevant comment. In response, we have calculated the WAF from the EAR5 dataset (for the period of 1940 to 2024), and added a new figure, Fig S3, to compare the simulated WAF for PI and ERA5. A common feature between the PI simulation and ER5 data is the spatial pattern, showing positive WAF over Eurasia and negative values over the N America continent. The main differences are the stronger WAF magnitudes of ERA5 data for both negative and negative values. We have mentioned this differences in text. Please note that we calculated the WAF for the PI climate state, rather than a composited WAF for weak SPV.

References:

Cohen, J., M. Barlow, P. Kushner, and K. Saito, 2007: Stratosphere-Troposphere coupling and links with Eurasian Land-Surface Variability, J. Climate, 20, 5335–5343.

Thanks for this reference, we have added it and discussed throughout the manuscript.